# Non-Destructive Characterization of Railway Materials and Components with Infrared Thermography Technique

**DOI:** 10.3390/ma12244077

**Published:** 2019-12-06

**Authors:** Jeongguk Kim

**Affiliations:** Advanced Railroad Vehicle Division, Korea Railroad Research Institute, Uiwang 16105, Korea; jkim@krri.re.kr; Tel.: +82-31-460-5518

**Keywords:** infrared (IR) thermography, non-destructive evaluation (NDE), railway materials, railway components, railway applications

## Abstract

Infrared (IR) thermography technology is one of the leading non-destructive evaluation (NDE) techniques based on infrared detection. Infrared thermography, in particular, has the advantage of not only being used in non-contact mode but also provides full images, real-time inspection, and relatively fast results. These advantages make it possible to perform thermal imaging analysis of railway materials and/or components, such as brake disc simulation, monitoring of abnormal heat generation, and monitoring of temperature changes, during mechanical tests. This study introduces the current state of research on railway materials and/or components using IR thermography technology. An attempt was made to characterize the deterioration of electrical equipment of diesel electric locomotives using infrared thermal imaging techniques. In addition, surface temperature monitoring was performed during tensile testing of railway steels using a high-speed infrared camera. Damage evolution due to the hot spot generation of railway brake discs was successfully monitored using high-speed IR cameras. In this paper, IR thermal imaging technology, used as a non-destructive evaluation analysis in the railway field, was introduced, and the results of recent research are presented.

## 1. Introduction

With the increasing speed of railway vehicles (to date, the maximum speed of commercial railway vehicles reaches up to 350 km/h), securing the driving safety of railway vehicles is becoming more important. In addition, the safety of railway components that ensure stable driving is becoming an increasingly important issue as well. In order to operate a safe railway vehicle, it is directly related to the maintenance activities of the railway system, and, accordingly, efficient maintenance is required. In general, many efforts have been made in the railway vehicle industry to manage various defects which are formed during operation or generated during manufacturing [1,2,3,4,5].

In the railway sector, various types of inspection methods have been used in terms of defect management. Various types of non-destructive evaluation (NDE) technologies are used during the regular maintenance of railway vehicles including ultrasonic testing (UT), magnetic particle testing (MT), radiography (RT), and infrared (IR) thermography [1,2,3,4,5]. Among the possible NDE technologies, infrared (IR) thermography is a relatively new technology in the field of railway maintenance and is an advanced non-destructive evaluation technique based on infrared detection [5,6,7,8,9,10]. In particular, infrared thermography technology is an advantage, providing fast full field and real-time inspection as well as contactless mode. These advantages enable thermal imaging analysis of various railway components such as temperature monitoring of railway brake disc surfaces, abnormal heating observations, and mechanical test monitoring, etc.

This study introduces recent research activities on IR thermography technology applied to railway materials and/or components. For example, the electric equipment of diesel electric locomotives was evaluated for degradation using infrared thermal imaging technology, and thermal characterization was performed. A high-speed infrared camera was used to measure surface temperature changes during tensile testing of railway steel materials and to investigate their mechanical properties and relationships. In addition, damage due to the fact of hot spot generation of railway brake discs was successfully monitored using a high-speed infrared camera and simulated braking characteristics. This paper introduces the IR thermography technique which is useful in the railway field and discusses recent research results.

Thus, the main objectives of this study on the NDE of railway materials and components using IR thermography technology were to (1) introduce the current state of railway field applications using infrared thermography technology, (2) explore the failure mode and fracture mechanism of railway steel materials with the help of IR thermography technology, (3) monitor the temperature changes during brake operation of railway brake discs, (4) perform thermal imaging inspections on electrical devices of diesel electric locomotives, (5) conduct thermographic detection on railway bogies using infrared thermography technology, and (6) develop non-destructive evaluation tools for detecting defects in railway vehicle components.

## 2. Experimental Procedures

In this paper, as discussed in the objectives of this study, several types of IR thermographic applications are introduced in the railway field. Therefore, in order to introduce these research activities and results, it is necessary to explain specific experimental conditions. The experimental procedures of this study are as follows.

### 2.1. Tensile Testing Interpretation of Railway Materials Using IR Thermography

In this study, two types of railway materials were used: axle railway steel for diesel electric locomotives and glass fiber-reinforced epoxy polymer matrix composites (PMCs) for railway bogies. In the case of railway steel materials, the tensile specimens for this study were prepared by machining on the axle of a diesel electric locomotive [6,7], and dog-bone flat specimens were prepared from the actual axle. For PMC samples, the actual tensile specimens were obtained from E-glass fiber-reinforced epoxy composites. Rectangular samples were prepared for the tensile test, and, because the material was PMC, steel tabs were attached in accordance with ASTM (American Society for Testing Materials) guidelines to avoid premature rupture of the grip portion that may occur when loading the specimen into the material testing machine [7].

Tensile tests were performed under load control at room temperature according to the Korean Standards (KS) guideline [7]. During the tensile test, a high-speed infrared (IR, A6700, FLIR, Santa Barbara, USA) camera monitored the surface temperature of the sample in order to investigate the failure mechanism and mode of failure. After the final fracture of the specimen, thermal image analysis was performed based on the thermal images obtained by IR thermography technology [7].

The camera’s maximum speed was up to 380 Hz at a 320 × 256 pixel size, and the camera’s maximum speed could be extended up to 20,000 Hz at 64 × 8 pixels in the reduced-resolution focal plane array. The spatial resolution of the camera was approximately 5.4 µm, and the temperature resolution was sensitive enough to achieve 0.02 °C at room temperature. The speed of the IR camera used in this study was 100 Hz for all samples, which means that a thermal image of 100 frames per second could be obtained. High-speed IR cameras ensure high-quality thermal imaging as well as instant image capture during tensile testing of the specimen [7].

### 2.2. Brake Disc Surface Monitoring during Braking Using Dynamometer

To determine the maximum surface temperature of railway brake discs for the purpose of safety, the surface temperature of the brake disc must be monitored during braking. Thermocouples are now used to measure this temperature and are mechanically measured by inserting a hole in the back of the disc. Infrared thermography, utilizing high-speed infrared cameras, provides a non-contact mode which is a useful way to monitor the surface temperature of the disc during braking. Braking tests were performed using brake discs and pads in accordance with KRS BR 0007 (non-asbestos disc brake pads) in accordance with the Korean Railway Standards (KRS) guidelines. The braking force was 25 kN on both sides. The initial braking speed (km/h) and the sequence of braking tests were as follows: 65, 35, 95, 65, 110, 95, 95, 35, 110, 65, 65, 95, 110, 110, 35, 35, and 65 km/h. Braking tests for this study were performed using a full-scale dynamometer operated at the Korea Railroad Research Institute (KRRI) [6,11,12].

### 2.3. Thermographic Inspection of High-Voltage Cables in Diesel Locomotive

Korea has enacted and implemented the Korea Railway Safety Law (KRSL) to ensure the performance and safety of railway vehicles. According to the KRSL, the life span of most railway vehicles is defined as 30 years. If a railway operator needs to extend the life of the vehicle, it should be diagnosed with items such as the current condition assessment, safety evaluation, and residual life assessment of the vehicle. According to the diagnosis results, it is legally possible to extend the service life up to 5 years [5].

In this study, as part of the precision inspection of railway vehicles, the insulation resistance measurement, degradation test, and deterioration evaluation of the electric equipment in diesel electric locomotives used for more than 30 years were carried out using infrared thermography [5]. 

A portable IR camera was used to perform abnormal temperature checks due to the presence of cable degradation and/or deterioration. The high-voltage cables were analyzed using IR cameras, in particular, the aging and current appearance of the high-voltage cables, connection areas, and terminal points [5].

### 2.4. Lock-In Thermographic Evaluation in Railway Bogie Frames

In this study, defect evaluation of railway vehicle bogies was performed by applying lock-in thermography, as an NDE technique. The railway vehicle bogies used in this study were applied to two types of bogies: polymer composite bogie and steel bogie [10].

The IR thermographic inspection described previously can be classified as passive thermography techniques. However, since active IR thermography works with other sources, such as external heating or ultrasonic vibrations, lock-in thermography can be described as an active infrared thermography technique (A6700, FLIR, Santa Barbara, USA). The principle of lock-in thermography is based on the synchronization of the heat source with the infrared camera. The heat source can be optical excitation, ultrasound, and periodic loading to the material. If the specimen is periodically loaded and subjected to heat waves, the resulting oscillating temperature field in the static region is recorded remotely via thermal infrared radiation. The frequency of modulation depends on the nature, size, and shape of the defect to be detected. This method reduces the effects of emissivity and non-uniform heating on temperature measurements, allowing inspection of a wide range of samples with high repeatability and sensitivity [10].

## 3. Results and Discussion

### 3.1. Tensile Testing Interpretation of Railway Materials Using IR Thermography

Figure 1a,b show the results of a full thermal image of the axle material on a diesel electric locomotive during the tensile test at 10 s before and at break, respectively. Figure 2 shows the stress–strain behavior of the tensile test on the diesel electric locomotive axle and wheel materials. The temperature changes during the tensile test are shown in Figure 3.

In Figure 1a,b, it can be seen that there were no significant differences in the surface temperature of the two tensile specimens at break. This result means that the final breakdown occurred very smoothly without any signs before final failure as shown in Figure 2. As shown in Figure 3, the final fracture shows an abrupt peak of temperature rise at break and occurs instantaneously. This means that the temperature rise peak at break was observed due to the specimen failure at the final failure of the specimen. These results indicate that IR thermal imaging can be a useful tool for analyzing the tensile failure behavior of railway axle materials.

Figure 4a,b show infrared thermal images taken every 0.01 seconds at the final fracture of the polymer composite specimen. Since the material used in this study was an E-glass fiber-reinforced epoxy polymer matrix composite (PMC), it was newly used as a bogie frame material for the weight lightening of railway vehicles in railway applications. As can be seen from the stress–strain behavior of the PMC material (Figure 5), the results of the tensile test indicate a brittle fracture mode without showing a plastic deformation area. The stress–strain results can be compared with the IR thermal images as shown in Figure 4a,b. The IR camera speed was 100 Hz, which means that each thermal image was obtained every 0.01 seconds. The immediate (within 0.01 second) and brittle form of failure was well documented by looking at the IR thermal image. In Figure 4b, some of the debris from the epoxy matrix illustrates how the final breakage was being made.

### 3.2. Brake Disc Surface Monitoring during Braking Using a Dynamometer

Figure 6 shows the results of the railway brake disc temperature monitoring using thermocouples and instantaneous friction coefficient values. In Figure 6, the surface temperature was measured by thermocouples embedded below the disc surface at different points. As the braking time increased, the temperature of the disc surface rose. However, the disc pad temperature remained almost constant due to the material properties of the non-asbestos disc brake pads.

Figure 7 shows the disc surface temperature monitoring results using an IR camera. As shown in Figure 7, the braking phenomenon was successfully described in terms of temperature increase. The braking start point was obtained in about 6 s and the braking was finished in about 21.4 s. In addition, the maximum temperature of the disc surface was shown in 11 s at about 170 °C (Figure 7). Figure 8 shows a typical thermal image when braking is started, at the highest temperature, and when braking is stopped. Hot spots were observed at peak temperatures as shown in Figure 8.

Inferring these results, it can be seen that high-speed infrared cameras can be useful tools for the simulation of braking phenomena and qualitative temperature monitoring and analysis of braking disc surfaces in the monitoring of railway braking tests.

### 3.3. Thermographic Inspection of High-Voltage Cables in Diesel Locomotives

Figure 9 and Figure 10 show a demonstration of the precision inspection of high-voltage cables under the Korean Railroad Safety Law as described in the Experimental Procedures (Section 2). Infrared cameras were used to check for rapid temperature changes or hot spots on the high-voltage cables connected to the traction motors in diesel electric locomotives. The high-voltage cable test was done with the power fully applied. Fortunately, no items, such as the special abnormal heat, or apparent abnormalities were found when inspecting the high-voltage cables (Figure 9 and Figure 10). The thermal imaging results of the high-voltage cable of the traction motor are shown by comparing the thermal image with the digital camera image. In the thermal image, as shown in Figure 9b and Figure 10b, the surface temperatures of the cables were 27.7 °C and 24.2 °C, respectively.

### 3.4. Lock-In Thermographic Evaluation in Railway Bogie Frames

Figure 11 shows the results of lock-in thermography and evaluation for railway bogies using E-glass fiber-reinforced epoxy polymer matrix composites. The conditions for performing lock-in thermography were as follows. Two halogen lamps (500 W) were applied as an external heat source, and the experiment was conducted by periodically applying heat to the specimen surface using a function generator. The excitation frequency used was in the range 0.1–0.01 Hz, and the optimum frequency was found. After applying the halogen heat source periodically, the surface defects of the PMC bogie were successfully displayed in phase, temperature, and amplitude as shown in Figure 11.

The optimum frequency for the lock-in thermographic image obtained in this study was 0.07 Hz. Similar experiments were applied to railway steel bogies that could be applied to commercial railway passenger cars. The results are shown in Figure 12. In Figure 12, abnormal areas were detected in the 0.06 Hz phase image, and these types of defects were found in the weldments on the bogies. Further investigation is likely to be needed for further study. Both temperature and amplitude thermal images show similar aspects.

### 3.5. Remarks on Infrared Thermography Applications in Railway Materials/Components Research

The concept of IR thermography began in the early 1950s, but IR technology is still unfamiliar within the railway industry. To date, thermal monitoring has been carried out primarily to identify thermal defects in railway components. However, as demonstrated in this paper, infrared thermography technology can be a powerful technique for explaining the failure of railway materials and/or component elements. Active and passive thermography can also be useful for finding new areas of interest. The examples of the various research activities described above should be the starting point for railway research, and the authors believe that more research efforts are needed in the railway sector.

## 4. Conclusions

The study on the non-destructive characterization of railway materials and components using infrared thermography has led to the following conclusions. Current applications in the field of railways were introduced to encourage widespread application of infrared thermography technology. Tensile failure behavior of axle materials and polymer matrix composites in railway vehicles was studied with the aid of infrared thermal imaging NDE technology. Temperature monitoring results during brake operation on railway brake discs were introduced, and high-speed infrared cameras were used to measure the surface temperature of the brake discs and for field monitoring of hot spot evolution. Infrared lock-in thermography technology provided a qualitative non-destructive tool for evaluating the integrity of railway bogies. Infrared thermography technology can be a reliable way to analyze deformation and/or failure analysis through temperature monitoring of railway components. In addition, both active and passive IR thermography can be useful for finding new areas of interest. The examples of the various research activities described above should be the starting point for railway research, and further research efforts are needed to make IR thermography technology more applicable to the railway field.

## Figures and Tables

**Figure 1 materials-12-04077-f001:**
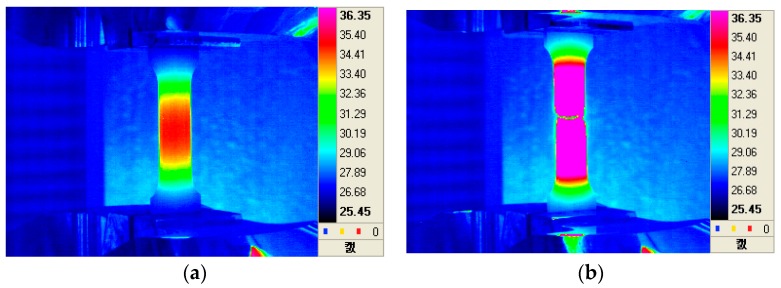
The thermographic images of axle steels in diesel electric locomotives during tensile tests: (**a**) at just 10 seconds before fracture and (**b**) at the time of failure.

**Figure 2 materials-12-04077-f002:**
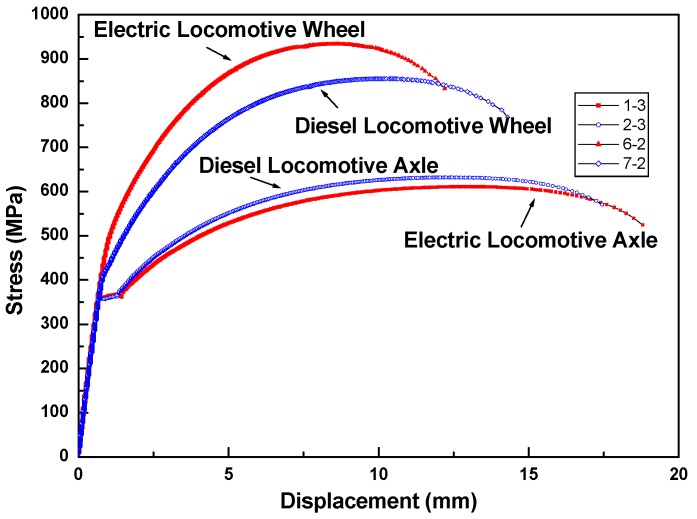
The stress–strain behavior during the tensile tests of the axle material.

**Figure 3 materials-12-04077-f003:**
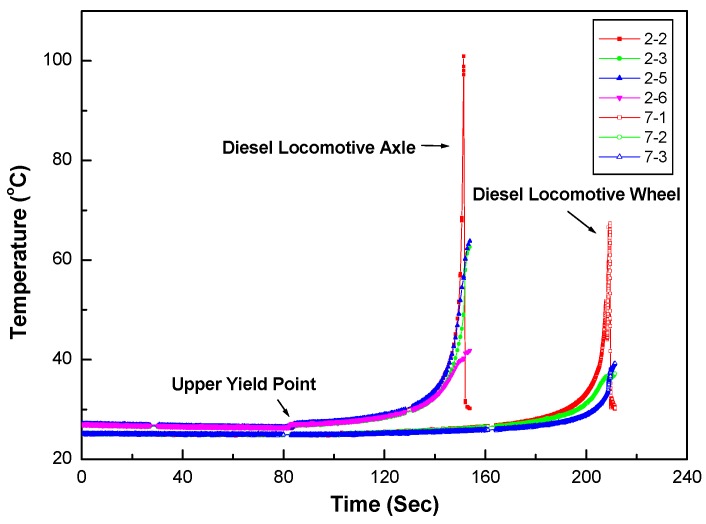
The temperature changes in the railway steel specimens’ surfaces during tensile test.

**Figure 4 materials-12-04077-f004:**
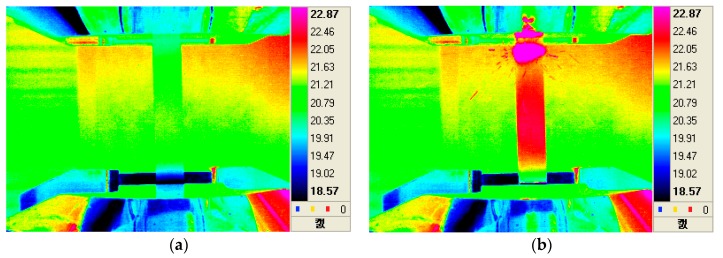
The infrared thermographic images during tensile test: (**a**) at 0.01 second before failure and (**b**) at the final failure.

**Figure 5 materials-12-04077-f005:**
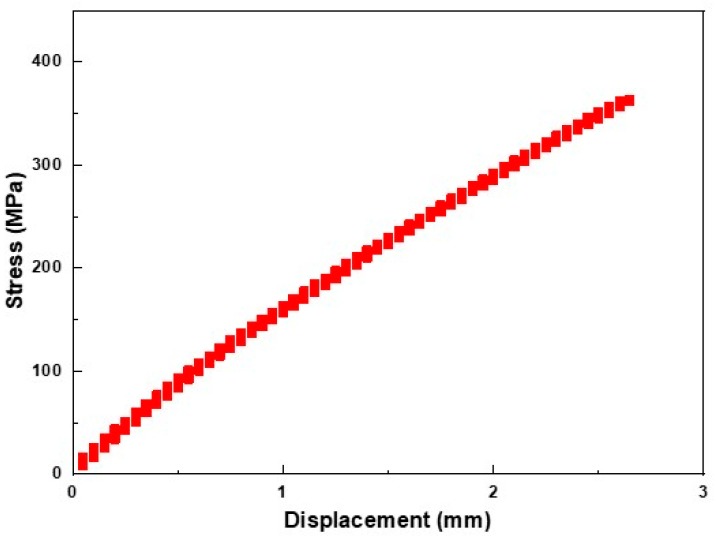
The stress–strain behavior of the polymer matrix composites.

**Figure 6 materials-12-04077-f006:**
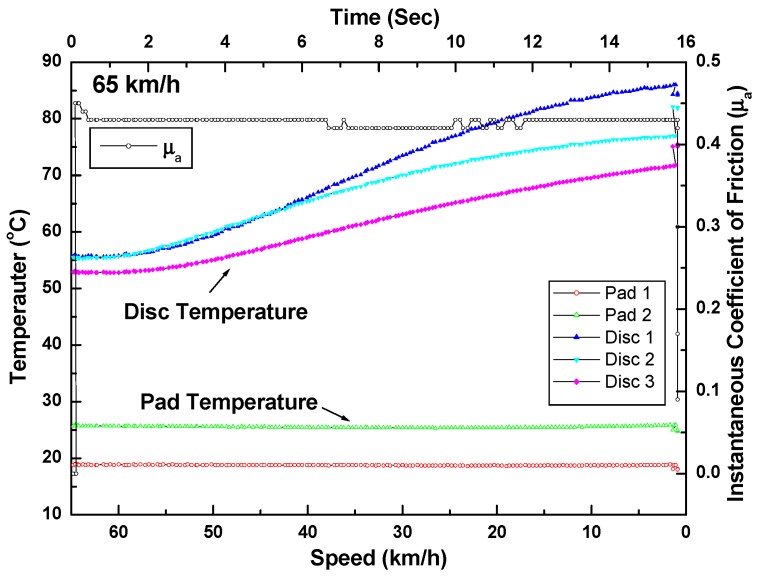
The brake disc temperature evolution using thermocouples with instantaneous coefficient of friction. Note that the initial braking speed was 65 km/h.

**Figure 7 materials-12-04077-f007:**
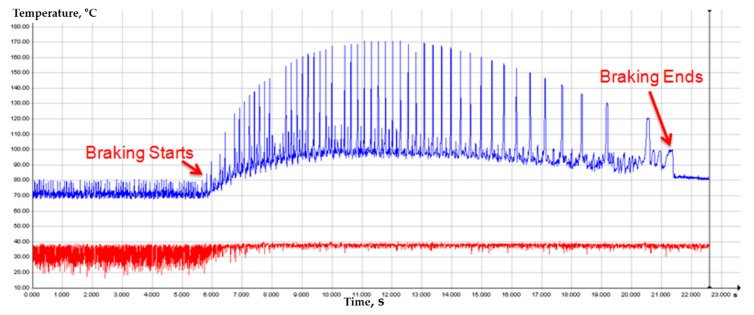
The disc surface temperature monitoring results using a high-speed IR camera.

**Figure 8 materials-12-04077-f008:**
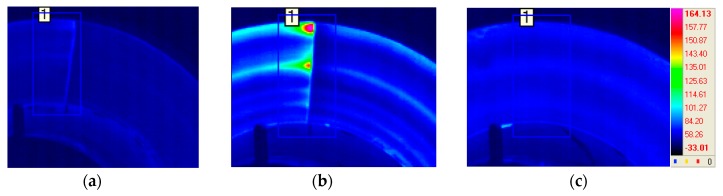
Infrared thermographic images: (**a**) at the beginning of braking, (**b**) at the peak temperature, and (**c**) at the time of braking stop.

**Figure 9 materials-12-04077-f009:**
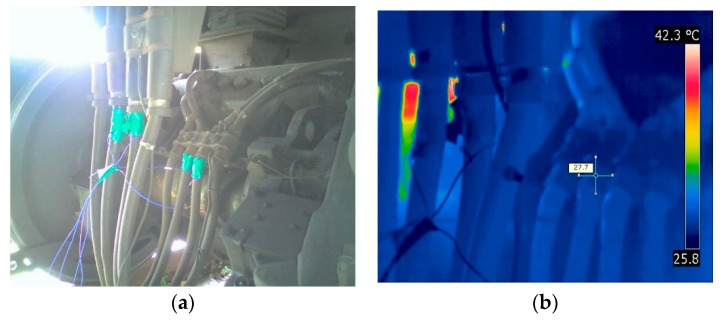
The thermographic inspection results of high-voltage cables in traction motor with (**a**) digital camera images and (**b**) IR thermographic images.

**Figure 10 materials-12-04077-f010:**
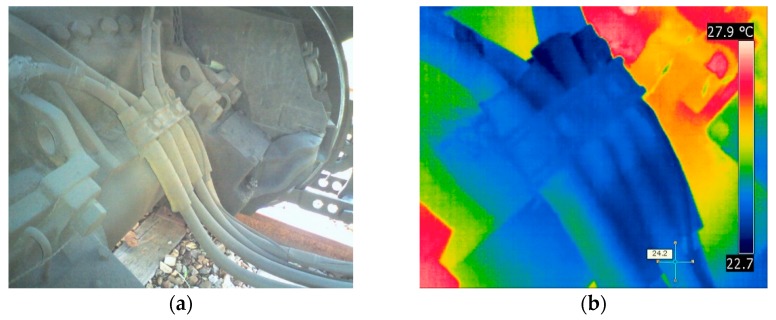
The thermographic inspection results of high-voltage cables in traction motor with (**a**) digital camera images and (**b**) IR thermographic images.

**Figure 11 materials-12-04077-f011:**
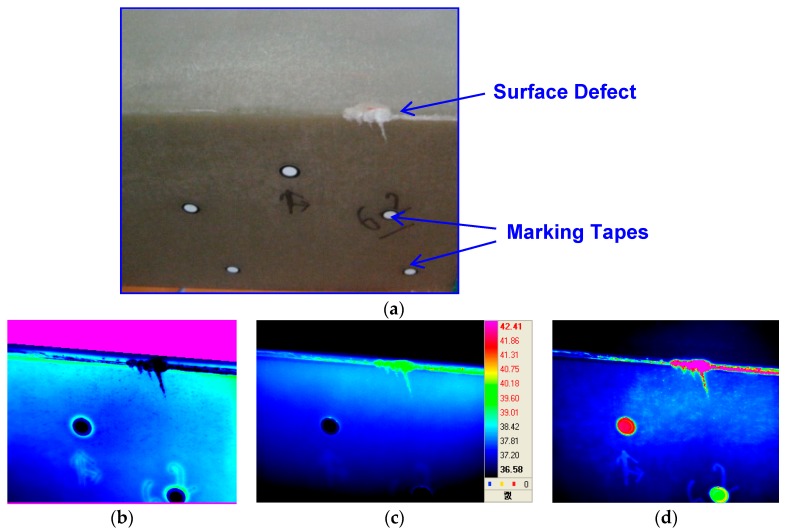
The lock-in thermography evaluation results for railway bogie with E-glass fiber-reinforced epoxy polymer matrix composites: (**a**) actual image, (**b**) phase IR image, (**c**) temperature IR image, and (**d**) amplitude IR image.

**Figure 12 materials-12-04077-f012:**
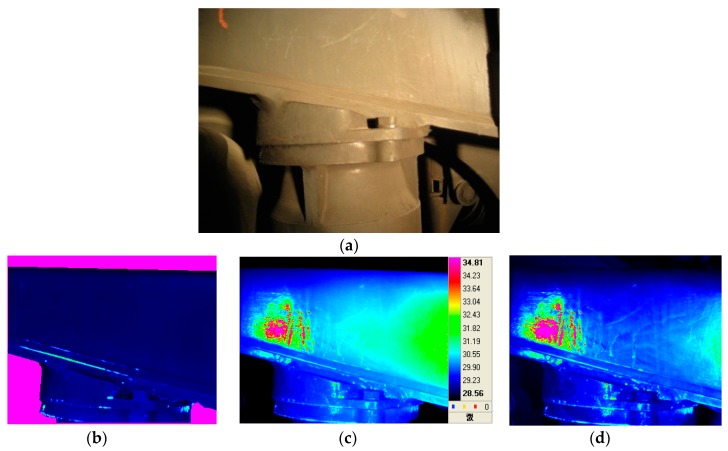
The lock-in thermography evaluation results for railway steel bogie: (**a**) actual image, (**b**) phase IR image, (**c**) temperature IR image, and (**d**) amplitude IR image.

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
