# Peer review of "Non-Destructive Characterization of Railway Materials and Components with Infrared Thermography Technique"

_materials, 2019, doi:10.3390/ma12244077_

Round 1

Reviewer 1 Report

The infrared thermography (IR) is proved to be a useful method in nondestructive evaluation of components. The subject is interesting, however there is nothing specified about the material constants needed in the performing IR evaluations or description of the concept method.

Author Response

Response to Reviewer 1 Comments

Point 1: The infrared thermography (IR) is proved to be a useful method in nondestructive evaluation of components. The subject is interesting, however there is nothing specified about the material constants needed in the performing IR evaluations or description of the concept method.

Response 1: Thank you for your comment on this article. As pointed out, experimental conditions for IR evaluation were introduced in Section 2.1, and new conditions for lock-in thermography were added in Section 3.4.

Thank you.

Reviewer 2 Report

The article should be supplemented with the following information:
- halogen lamp power in lock-in tests,
- temperature scale in figures 8, 11c and 12 c.

Author Response

Response to Reviewer 2 Comments

Point 1: The article should be supplemented with the following information:

- halogen lamp power in lock-in tests,

- temperature scale in figures 8, 11c and 12 c.

Response 1: Thank you for your comment on this article. As pointed out, we provided halogen lamp power information (500W * 2) in Section 3.4, and temperature scales in Figures 8, 11c and 12c were added.

Thank you.
